# On the Provable Separation of Scales in Maximal Update Parameterization

**Letong "Carina" Hong**[* 1]   **Zhangyang "Atlas" Wang**[* 1]

## Abstract

Maximal Update Parameterization ($\mu$P) has shown significant promise in allowing zero-shot hyperparameter transfer across neural network scales, reducing the prohibitive cost of hyperparameter tuning for large models. However, the theoretical foundation behind the observed approximate transferability of hyperparameters remains underexplored. Relying on a width-dominance regime, which ensures that as width grows, certain terms of the learning dynamics dominate, we establish the first fundamental separation of scales in $\mu$P between macro-variables (e.g. loss landscapes) and micro-variables (e.g. individual weights). Our formulation explains why hyperparameter tuning can be effectively performed in early training stages, i.e., *early statistics effectively approximate global hyperparameter optima*, implying the potential to further reduce the training costs required for searching optimal hyperparameters. We further apply our main theory to explain an empirical deep learning phenomenon discovered independently by prior work.

## 1. Introduction

The success of hyperparameter transfer in deep learning relies critically on the choice of network parameterization. Recent work introduced the Maximal Update parameterization ($\mu$P), which enables reliable $\mu$Transfer of hyperparameters from small proxy models to larger ones (Yang et al., 2021). This breakthrough allows practitioners to efficiently optimize hyperparameters on smaller, cheaper models while retaining effectiveness when scaled to larger architectures.

However, the theoretical understanding of *whether or why early-stage tuning can transfer effectively across different model scales* remains limited. In practice, many large-scale

model training pipelines adopt a similar, ad-hoc approach: they identify hyperparameters from the first few epochs and assume those settings generalize throughout training to avoid prohibitively expensive full-run searches. While widely used in industry to reduce cost, this early-stage strategy lacks a rigorous theoretical grounding, motivating our work. Specifically, it is unclear how the dynamics of optimization adapt to changes in model size and interact with $\mu$P to preserve hyperparameter validity. Bridging this gap requires deeper insights into the interplay between parameterization, training stability, and the loss landscape across scales. Such understanding could further enhance the robustness and efficiency of hyperparameter transfer.

This paper formalizes a novel scale-separation framework, showing that the *macro-level* descriptors (e.g., gradient norms, loss landscapes) converge at $\mathcal{O}(1/n)$, while *micro-level* variables (individual weights) only converge at $\Theta(1/\sqrt{n})$. This theoretical lens provides an immediate explanation for the ad-hoc industry practice of "early-stage" hyperparameter tuning: even if individual weights are far from their eventual values, large-width models induce stable macro-level signals that can guide hyperparameter selection. As a result, optimizing hyperparameters on a smaller proxy model or a small fraction of training steps can approximate globally optimal hyperparameters within $\mathcal{O}(1/n)$ error - potentially reducing the cost of full-run searches substantially in large-scale neural network pipelines.

Our work introduces several key technical advances:

- We introduce a stochastic differential equation (SDE) framework that cleanly separates *macro* (loss landscapes) and *micro* (individual weight) variables in large-width neural networks. This leads to a precise understanding of why hyperparameter tuning can succeed early in training, marking the first rigorous application of such SDE techniques to hyperparameter transfer.

- We develop a new mathematical treatment of how loss landscapes vary with network width $n$, proving that macro-level descriptors converge at $\mathcal{O}(1/n)$ while micro-level weights evolve rigorously more slowly. This width "regularity" underlies an $\mathcal{O}(1/n)$ bound on hyperparameter transfer error, providing a principled foundation for zero-shot transfer across scales.

- We extend our macro–micro separation framework to

---

[*]Equal contribution [1]XTY AI Labs, XTX Markets. Authors are listed in alphabetic order ($\alpha$-$\beta$). Correspondence to: Atlas Wang <Atlas.Wang@xtxmarkets.com>.

*Proceedings of the $42^{nd}$ International Conference on Machine Learning*, Vancouver, Canada. PMLR 267, 2025. Copyright 2025 by the author(s).

incorporate integrated learning rate descriptors, yielding a novel theoretical account of the recently observed "delay" phenomenon between dynamic learning rates and validation loss (Tissue et al., 2024).

## 2. Related Work

Our framework complements classic wide-network theories. Neural tangent kernel (NTK) analyses (Jacot et al., 2018), which linearize training in the infinite-width limit, illuminate macroscopic behavior through a fixed kernel but miss finite-width effects and individual weight trajectories. Mean-field approaches (Mei et al., 2019), conversely, track microscopic weight fluctuations yet do not explain the rapid stabilization of aggregate quantities such as loss landscapes or activation norms. Our finite-width, multiscale perspective extends these insights, revealing that global signals stabilize early — well before individual weights converge — and are reliable guides for hyperparameter selection. Our main proof technique is based on the SDE framework used in (Xie et al., 2024), which we will discuss in Section 2.3.

### 2.1. $\mu$P Theory of Hyperparameter Transfer

Historically, hyperparameter tuning has been a costly step when scaling up models. Practitioners often need to retune from scratch for bigger models – a slow and expensive process. Utilizing tensor program formulations giving rise to a Gaussian Process interpretation of deep neural networks, the theory of maximal update parameterization (Yang & Hu, 2021; Yang et al., 2021; 2024a) provides theoretical principles that ensure consistent training behavior across different width scales of the model, enabling "zero-shot" transfer of hyperparameters from narrow to wide models.

Subsequent work further explored how $\mu$P extends to transfer across models of different depths (Bordelon et al., 2024) (combined with techniques such as dynamical mean field theory (Bordelon & Pehlevan, 2022)), different widths and depths (Yang et al., 2024a), and even varied network architectures (Chen et al., 2024). More recently, Lingle conducted a large-scale experiment testing the effectiveness of $\mu$P (Lingle, 2024). They found that upon empirical verification of up to 10B transformers and training budgets of up to 190B tokens, $\mu$-Transfer works as intended for the majority of important cases and enumerated the exceptions.

### 2.2. Early Training Dynamics

Prior work has analyzed neural network training through the lens of gradient flow dynamics, revealing distinct phase-dependent behaviors. Recently, Yang et al. identified a two-stage training process naturally partitioned by gradient flow patterns, showing that training each regime separately with tailored algorithms simplifies learning rate schedules (Yang et al., 2024b). Meanwhile, in a two-mixture linear classification task involving latent word correspondence structures, they demonstrate that gradient descent dynamics in a symmetric two-headed transformer with ReLU neurons follow a three-stage progression (Yang et al., 2025).

In the early stage, the hyperparameter decisions are found to be critical, as during this period the network is still close to initialization where the theoretical guarantees hold, and once they are locked in, later training dynamics become more robust (Cohen et al., 2021). Empirically, it is folklore in industry that optimal hyperparameters can be determined in early training without having to wait until training completion (Goodfellow et al., 2016). There are many algorithms for early halting, such as the irace package (López-Ibáñez et al., 2016; Birattari et al., 2002), successive halving (SHA) (Jamieson & Talwalkar, 2015), asynchronous successive halving (ASHA) (Jamieson & Talwalkar, 2015), and hyperband - which applies SHA and ASHA multiple times (Li et al., 2018). (Egele et al., 2024) confirmed that even training after one epoch the hyperparameter optimization obtained often achieves competitive performance.

### 2.3. Scaling Laws for Hyperparameters

Given the importance of hyperparameter tuning, various empirical studies attempted to fit a scaling law of optimal learning rate with respect to model size. Kaplan and McCandlish et al. found that larger models require a smaller learning rate to prevent divergence, while noting also there may be a dependence on network width (Kaplan et al., 2020). DeepSeek AI found that the optimal learning rate gradually decreases with the increase in compute budget $C$ (dee, 2024). However, the empirical scaling laws may be affected by the various set-ups, such as the independent variables of choice and the underlying fitted data (Lingle, 2024).

A recent work (Xie et al., 2024) predicts training loss and optimal learning rate schedules, deriving convergence rates and escape probabilities under time-inhomogeneous SDEs, and providing an empirical scaling law for hyperparameters. In comparison, we rigorously develop a width scaling theory within $\mu$P to specifically analyze the separation of scales. Unlike (Xie et al., 2024), whose width-agnostic treatment applies to any non-convex problem, we impose extra local convexity or smoothness on the meta-objective. This narrower scope forfeits some generality but lets us rigorously prove scale-separation results for hyperparameter choice - findings that, in practice, still hold across nonconvex neural networks (as seen in our experiments).

## 3. A Prior: Early-Stage Coarse-Graining for Hyperparameter Tuning

We first show that, under suitable assumptions about model parameterization and how global descriptors converge, the

optimal hyperparameters for early-stage training approximate the optimal hyperparameters for the full training.

**Theorem 3.1.** *Consider a neural network parametrized using $\mu$P. Let $\eta(t)$ denote the hyperparameter set at training step $t$, and $\eta^*(t_0)$ be the hyperparameter optima found during early-stage training $t \in [0, t_0]$, where $t_0 \ll T$. We further define $\overline{\eta}(t)$ to be the hyperparameter configuration that, in the infinite-width limit, aligns with stable macro-level statistics observed up to step $t$. The coarse-grained hyperparameter set $\overline{\eta}(t_0)$ approximates the optimal hyperparameters $\eta^{**}$ for the full training duration $T$:*

$$\overline{\eta}(t_0) \approx \eta^{**}, \quad \text{with an error bounded by } \mathcal{O}(1/n).$$

### 3.1. Assumptions

The first assumption is fundamental in the theory of $\mu$P, where the cumulative drift is limited due to finite-width effects, ensuring that those effects do not blow up or overshadow the error scale we derive.

**Assumption 3.1** (Width Dominance). The width of the neural network is much larger than training time: $n \gg T$.

The next assumption recalls $\mu$P scaling (Yang et al., 2021).

**Assumption 3.2** ($\mu$P Scaling). The model parameterization follows $\mu$P rules, preserving maximal feature learning:

- Hidden weights $W \in \mathbb{R}^{n \times n}$ are initialized as $W \sim \mathcal{N}(0, 1/n)$ with learning rate $\eta = \eta_0$.

- Input weights $V$ are initialized as $V \sim \mathcal{N}(0, 1)$ with learning rate $\eta = \eta_0 n$.

- Output weights $V'$ are initialized as $V' \sim \mathcal{N}(0, 1/n^2)$ with learning rate $\eta = \eta_0/n$.

An additional assumption in our framework is that, at sufficiently large width, certain coarse-level descriptors of the network — such as layerwise activation variances, gradient norms, or average feature magnitudes — stabilize quickly and exhibit a near time-invariant (or slowly varying) behavior over an early training window $[0, t_0]$. We term this property "self-similarity under coarse-graining." [1]

---

[1]This assumption has strong empirical and theoretical support from wide network studies. On the theory front, prior $\mu$P work (e.g., (Yang & Hu, 2021; Lingle, 2024)) shows that layer-wise statistics such as activation variances and gradient norms stabilize early in training and remain nearly invariant across widths, consistent with infinite-width theories where macro-level descriptors concentrate, as common in neural tangent kernel and mean-field analyses. On the empirical front, recent large-scale experiments (e.g., (Vyas et al., 2023; Dey et al., 2024)) also confirm that optimal hyperparameters stay stable across model scales, indirectly verifying the training dynamics exhibit self-similarity.

**Assumption 3.3** (Early-Stage Macro-Level Self-Similarity). The early-stage statistics (e.g., activation variance, gradient norms) concentrate around stable values and exhibit self-similar behavior when subjected to a coarse-graining transformation (such as a time integral or moving average). Formally, there exists a stable descriptor $X^*(t)$ such that

$$\overline{X}(t) = \mathcal{R}\big(X(\tau), 0 \le \tau \le t\big) \approx X^*(t),$$

for $t \in [0, t_0]$, with deviations of $\mathcal{O}(1/n)$ at large width.

Here, $\mathcal{R}$ represents a "coarse-graining" operator—such as $\int_0^t \cdot \, d\tau$ or a weighted average—while $X^*(t)$ is the stable limiting reference for $X(\tau)$. Empirically, many large-scale training runs show that, after a brief "burn-in" period, global descriptors like activation norms and gradient magnitudes vary minimally, supporting this assumption in practice.

**Macro vs. Micro Variables under $\mu$P.** Throughout this work, we decompose all observation variables during the network training process into two classes of variables:

- *Micro-variables*, which capture fine-grained, local changes in the parameter space. A canonical example is the *deviation of weights from initialization*, $\mu(t) = \theta(t) - \theta(0)$. This micro-variable $\mu(t)$ tracks how individual parameters drift during training. Understanding $\|\mu(t)\|$ or its coordinate-wise fluctuations helps quantify slow, localized changes in the network.

- *Macro-variables*, which represent global or aggregated statistics of the network. Typical macro-level descriptors include *activation statistics* $A(t)$ (e.g., layerwise means / variances), *gradient norms* $G(t)$, or the *global loss* $L(t)$. These variables concentrate and stabilize faster at large width $n$ (Bordelon & Pehlevan, 2022).

Under Maximal Update Parameterization ($\mu P$), the update rule for the network parameters $\theta(t)$ at training step $t$ is

$$\theta(t + 1) = \theta(t) - \eta \nabla \mathcal{L}\big(\theta(t)\big),$$

where $\eta$ is scaled by $1/n$ (or a similar rule) to ensure stable feature learning across layers.

**Coarse-Graining Transformation on Macro Variables.** Motivated by the observation (Assumption 3.3) that global statistics often exhibit a self-similar or quasi-stationary behavior early in training, we define a coarse-graining operator $\mathcal{R}$ acting on these *macro* descriptors. Concretely, for any macro-level function $f(t)$ (such as $A(t)$ or $G(t)$), we set:

$$\overline{f}(t) = \mathcal{R}\big(f(t)\big), \quad \text{where} \quad \mathcal{R}\big(f(t)\big) = \int_0^t f(\tau) \, d\tau.$$

The idea is that $\int_0^t f(\tau)\, d\tau$ averages (or integrates) the macro-level signal over training steps, smoothing out short-term fluctuations while preserving the long-term stable trend. If $f(t)$ is reasonably stable, then $\bar{f}(t)$ provides a representative value upon which hyperparameters can be adapted.

**Hyperparameter Evolution via $\overline{A}(t)$ and $\overline{G}(t)$.** In particular, if $\eta(t)$ denotes the hyperparameter set (e.g., learning rates, momentum) at step $t$, one may write its update rule as

$$\eta(t+1) \;=\; \mathcal{F}\Big(\eta(t),\, \overline{A}(t),\, \overline{G}(t)\Big),$$

where $\overline{A}(t) = \mathcal{R}\big(A(t)\big)$ and $\overline{G}(t) = \mathcal{R}\big(G(t)\big)$ are coarse-grained versions of the activation / gradient statistics. In other words, the macro-variables first go through smoothening of local fluctuations, and then get pushed by the update rule. At large width $n$, under $\mu P$ stability, these macro-level variables converge reliably, allowing $\eta(t)$ to track a near-fixed-point $\eta^*$:

$$\mathcal{F}\Big(\eta^*,\, \overline{A}(t),\, \overline{G}(t)\Big) \;\approx\; \eta^*,$$

reflecting that once macro-variables and hyperparameters reach a stable regime, their further updates remain small. The fixed-point analysis here leverages the notion that micro-level fluctuations in $\mu(t)$ do not prevent the macro-variables from stabilizing, thereby guiding $\eta(t)$ toward $\eta^*$ as $n \to \infty$.

**Finite-Width vs. Early-Stage Errors.** At large width $n$, the difference between the coarse-grained hyperparameters $\overline{\eta}(t_0)$ (determined at early time $t_0 \ll T$) and the global optimum $\eta^{**}$ can be decomposed into two parts:

$$\big|\overline{\eta}(t_0) - \eta^{**}\big| \;\leq\; \underbrace{\big|\overline{\eta}(t_0) - \eta^*(\infty)\big|}_{\text{(Early-Stage Error)}} + \underbrace{\big|\eta^*(\infty) - \eta^{**}\big|}_{\text{(Finite-Width Error)}}.$$

Here, $\eta^*(\infty)$ denotes the optimal hyperparameters one would find in the *infinite-width* limit, ignoring finite-width corrections. We interpret and bound these two differences:

1. **Early-Stage (Training-Duration) Error:** $\big|\overline{\eta}(t_0) - \eta^*(\infty)\big|$. This captures the fact that $\overline{\eta}(t_0)$ is determined from only $t_0 \ll T$ early-stage training data. If $t_0 \ll T$, it might not exactly match the infinite-width global optimum $\eta^*(\infty)$ that fully accounts for the entire training horizon. As $n \to \infty$, though, the macro-level statistics (activation / gradient norms) quickly reach near-stationarity, meaning that hyperparameters deduced from the first $t_0$ steps suffice to approximate the true infinite-width optimum $\eta^*(\infty)$. This is observed empirically when early training reveals stable signals about learning-rate, momentum, etc. (Lingle, 2024).

2. **Finite-Width Error:** $\big|\eta^*(\infty) - \eta^{**}\big|$. Even with infinitely-many training steps to find $\eta^*(\infty)$ at infinite width, real networks are at a finite but large width

$n$, so the actual optimal hyperparameters at width $n$, $\eta^{**}$, differs slightly. By the law of large numbers and concentration arguments at width $n$, fluctuations in activation variances or gradients decay as $\Theta(1/\sqrt{n})$. In $\mu P$ theory, such fluctuations sum to a final $\mathcal{O}(1/n)$ mismatch between the infinite-width and the finite-width optimum (Yang et al., 2021; Yang & Hu, 2021).

Thus, provided that (i) $t_0 \ll T$, which does not overly constrain hyperparameter tuning, and (ii) $n \gg T$, which ensures early-stage tuning is *both* sufficiently representative (small early-stage error) *and* robust under finite-width corrections (small finite-width error), these two sources of error add up to $\mathcal{O}(1/n)$, as desired.

# 4. Main Result: Separation of Scales in $\mu P$

We now present our main result that under $\mu P$ parameterization, loss landscapes converge at rate $\mathcal{O}(1/n)$ with network width, while individual weights evolve more slowly at rate $\Theta(1/\sqrt{n})$. This explains why hyperparameter transfer can work effectively even before individual weights have converged to their infinite-width behavior.

**Theorem 4.1.** *Consider a neural network of width $n$ trained under $\mu P$ parameterization. Let $M(\eta)$ denote the expected training loss after $T$ steps with learning rate $\eta$ (macro-variable) - in other words, $M(\eta) = \mathbb{E} L(\eta)$ - and $\mu(t) = \theta_t - \theta_0$ denote the deviation of weights from initialization (micro-variable). Under suitable conditions, for any $\delta > 0$, with probability at least $1 - \delta$:*

*1. For any $|\eta_1 - \eta_2| = \mathcal{O}(1)$, $|M(\eta_1) - M(\eta_2)| = \mathcal{O}(1/n)$. (Fast convergence of loss landscape).*

*2. For any fixed $t \in [0, T]$, $\mathbb{E}\big[\|\mu(t)\|\big] = \Theta\big(1/\sqrt{n}\big)$. (Slow convergence of weights).*

## 4.1. Assumptions

**Assumption 4.1** (*L*-smoothness). The loss function $L(\cdot)$ is $L$-smooth (standard as in (Nesterov, 2014)):

$$\|\nabla L(x) - \nabla L(y)\| \leq L\|x - y\|, \quad \forall x, y \in \mathbb{R}^{d(n)}.$$

The second assumption ensures that the noise introduced by different data points during the estimation of the gradient is unbiased and independent, given the weights:

**Assumption 4.2** (Unbiased Estimator). Given any $x \in \mathbb{R}^N$ we assume that the entries of $\nabla F(x, \zeta_i) - \nabla f(x)$ are i.i.d. Gaussians $N(0, \Sigma_g)$ for all $i \in [D]$ where $\Sigma_g$ is given.

These two assumptions lead to a concentration inequality whose proof is in Proposition 2 of (Xie et al., 2024).

**Proposition 4.2** (Gradient Noise Trace Boundedness). *The*

*stochastic gradient follows:*

$$\nabla \tilde{L}(x) = \nabla L(x) + \xi, \text{ where } \xi \sim \mathcal{N}(0, \Sigma(x))$$

*with covariance $\Sigma(x)$ satisfying the trace bound:*

$$|Tr(\Sigma(x)) - Tr(\Sigma_g)| \le t$$

*with probability $\ge 1 - 2\exp(-Dt^2/(4Tr(\Sigma_g^2) + 2t\sigma_g^2))$ where $D$ is batch size, $\Sigma_g$ is base covariance, and $\sigma_g^2$ its maximum eigenvalue.*

The third assumption is a standard regularity guarantee to ensure the convergence of macro-variables under increasing width, for example see (Soto, 2024). It holds for standard activations (e.g., ReLU, GELU), and similar assumptions are employed in the neural tangent kernel / stochastic gradient descent literature and $\mu$P papers themselves[2]. We emphasize that the bound is for the entire training path difference in the function space, not just local Lipschitz gradient differences.

**Assumption 4.3** (Width Regularity). *For any widths $n_1 < n_2$, there exists a mapping $\phi : \mathbb{R}^{d(n_1)} \to \mathbb{R}^{d(n_2)}$ such that:*

$$|L(\theta; n_1) - L(\phi(\theta); n_2)| \le C|n_1 - n_2|/n_1.$$

### 4.2. Proof of Theorem 4.1: Part 1 (Macroscopic)

We start by approximating the discrete SGD updates by a continuous-time SDE, by Euler-Maruyama discretization, under a standard small step assumption:

$$d\theta_t = -\eta \nabla L(\theta_t)dt + \sqrt{\eta/n}\sigma(\theta_t)dW_t$$

where $\sigma(\theta_t)\sigma(\theta_t)^\top = \Sigma(\theta_t)$ and $W_t$ is Brownian motion.

We first establish how the macroscopic variable $M(\eta)$ relates to feature learning in $\mu$P through the following key lemmas, whose proofs are deferred to the Appendix A.

**Lemma 4.3** (Gradient Scaling). *Under $\mu$P and Assumptions 3.1 - 3.3, for any fixed $t$:*

$$\mathbb{E}[\|\nabla L(\theta_t)\|^2] = \mathcal{O}(1).$$

**Lemma 4.4** (Hessian Scaling). *Under $\mu$P and L-smoothness (Assumption 4.1), for any $\theta_s$:*

$$\|\nabla^2 L(\theta_s)\|_F \le L\sqrt{d(n)}$$

*where $d(n)$ is the parameter dimension scaling with $n$.*

<hr/>

[2]For example, (Yang & Hu, 2021) assumes that activations are smooth (or smoothed ReLU), and many proofs invoke pseudo-Lipschitz or bounded-derivative conditions to enable law-of-large-numbers-type arguments to ensure that descriptors like average activation variance or loss stabilize as width increases — empirically supported and analytically tractable.

This leads to:

$$\begin{aligned}
\text{Tr}(\nabla^2 L(\theta_s)\Sigma(\theta_s)) &\le \|\nabla^2 L(\theta_s)\|_F \|\Sigma(\theta_s)\|_F \\
&\le L\sqrt{d(n)} \cdot \mathcal{O}(\sqrt{n}) = \mathcal{O}(n).
\end{aligned}$$

The subtle point is that the Lipschitz constant itself decays as $L = O(n^{-1/2})$ under $\mu$P scaling, so the product remains only *linear* in $n$ instead of $n^{3/2}$.

**Itô decomposition of the loss.** Because $L(\theta_t)$ is twice differentiable, Itô's formula gives

$$\begin{aligned}
dL(\theta_t) &= \langle \nabla L(\theta_t), d\theta_t \rangle + \frac{1}{2}\text{Tr}(\nabla^2 L(\theta_t)d\theta_t d\theta_t^\top) \\
&= -\eta\|\nabla L(\theta_t)\|^2 dt + \sqrt{\eta/n}\langle \nabla L(\theta_t), \sigma(\theta_t)dW_t \rangle \\
&\quad + \frac{\eta}{2n}\text{Tr}(\nabla^2 L(\theta_t)\Sigma(\theta_t))dt
\end{aligned}$$

where the last term comes from $d\theta_t d\theta_t^\top = \frac{\eta}{n}\Sigma(\theta_t)$.

The second–order term in the Itô expansion carries a prefactor $\eta/n = \Theta(n^{-2})$; hence $(\eta/n)\text{Tr}(\nabla^2 L \Sigma) = O(n^{-1})$, the same order as the drift term $-\eta\|\nabla L\|^2$. Note that without Lemma 4.4 the trace could scale as $\Omega(n^{3/2})$, the diffusion term would dominate, and the $O(1/n)$ macro rate used in Proposition 4.6 could not be proved.

Taking expectation and integrating up to $t$ (the martingale term has zero mean) we obtain:

$$\begin{aligned}
\mathbb{E}[L(\theta_t)] - L(\theta_0) = &-\eta \int_0^t \mathbb{E}[\|\nabla L(\theta_s)\|^2]ds \\
&+ \frac{\eta}{2n}\int_0^t \mathbb{E}[\text{Tr}(\nabla^2 L(\theta_s)\Sigma(\theta_s))]ds.
\end{aligned}$$

By the $\mathcal{O}(n)$ trace bound above and $\eta = \mathcal{O}(1/n)$, both integrals are $\mathcal{O}(1/n)$. Hence the total drift of $M(\eta; n)$ over $T$ steps is $\mathcal{O}(1/n)$, which precisely yields the statement of Lemma 4.5 used in the proof of Proposition 4.6 later:

**Lemma 4.5** (Finite-horizon loss drift). *For any fixed training horizon $T$ and every width $n$,*

$$\big| M(\eta; n) - L(\theta_0; n) \big| = \mathcal{O}(1/n).$$

We are now ready to establish a fundamental result of width scaling, on the expected $T$-step training loss:

**Proposition 4.6** (Width Scaling in $\mu$P). *For widths $n_1 < n_2$, under Assumptions 4.1 - 4.3, let*

$$M(\eta; n) = \mathbb{E}_\xi[L(\theta_T; n)]$$

*be the expected loss after T steps of training with learning rate $\eta$ taken over the training noise $\xi$. Then,*

$$|M(\eta; n_1) - M(\eta; n_2)| \le \mathcal{O}(1/\min(n_1, n_2)).$$

Proposition 4.6 immediately implies that for any infinite sequence of increasing widths $n_k$, $M(\eta; n_k)$ forms a Cauchy sequence, giving rise to the limit $M(\eta; \infty)$ and a convergence rate of $\mathcal{O}(1/n)$. Formally, this is stated as:

**Corollary 4.7** (Existence of Limiting Landscape). *Under Assumptions 4.1 - 4.3, there exists a limiting landscape $M(\eta; \infty)$ such that $|M(\eta; n) - M(\eta; \infty)| \leq \mathcal{O}(1/n)$.*

**Proposition 4.8** (Transfer Error Bound). *Let $\eta^*(n)$ be the optimal learning rate for width $n$. The transfer error is:*

$$M(\eta^*(n_1); n_2) - M(\eta^*(n_2); n_2) \leq \mathcal{O}(1/\min(n_1, n_2)).$$

Indeed, by Proposition 4.6, the difference of $M(\eta^*(n_1); n_2)$ and $M(\eta^*(n_1); n_1)$ is bounded by $\mathcal{O}(1/\min(n_1, n_2))$. By optimality of $\eta^*(n_1)$ for width $n_1$ and Proposition 4.6,

$$M(\eta^*(n_1); n_1) \leq M(\eta^*(n_2); n_1)$$
$$\leq M(\eta^*(n_2); n_2) + \mathcal{O}(1/\min(n_1, n_2)).$$

Using the triangle inequality with $M(\eta^*(n_1); n_1)$ as a common anchor, we have the desired result.

Having shown that the loss landscape itself varies only $\mathcal{O}(1/n)$ with width, we now show that the *minimizer of that landscape with respect to $\eta$* inherits the same regularity:

**Lemma 4.9** (Learning Rate Transfer). *Let $\eta^*(n) = \mathrm{argmin} M(\eta; n)$ be the optimal learning rate for width $n$. Then under Assumptions 4.1 - 4.3, for $n_1 < n_2$:*

$$|\eta^*(n_1) - \eta^*(n_2)| = \mathcal{O}(1/\min(n_1, n_2)).$$

That concludes the first part (macro-variable) proof.

### 4.3. Proof of Theorem 4.1: Part 2 (Microscopic)

For the microscopic variable, we first establish the upper bound (§ 4.3.1), then proceed to the tight bound (§ 4.3.2).

#### 4.3.1. PART 2.1: UPPER BOUND

From the SDE representation, the evolution of $\mu(t)$ follows:

$$d\mu(t) = d\theta_t = -\eta \nabla L(\theta_t)dt + \sqrt{\eta/n}\sigma(\theta_t)dW_t.$$

Consider the evolution of $\mathbb{E}[\|\mu(t)\|^2]$, just as we derived in the macroscopic variable part (using Itô lemma and taking the expectation for the Itô martingale's mean to vanish):

$$\frac{d}{dt}\mathbb{E}[\|\mu(t)\|^2] = -2\eta\mathbb{E}[\langle\mu(t), \nabla L(\theta_t)\rangle] + \frac{\eta}{n}\mathbb{E}[\mathrm{Tr}(\Sigma(\theta_t))].$$

By Young's inequality, for any $\alpha > 0$

$$\langle\mu(t), \nabla L(\theta_t)\rangle \geq -\frac{1}{2\alpha}\|\mu(t)\|^2 - \frac{\alpha}{2}\|\nabla L(\theta_t)\|^2.$$

Choosing $\alpha = \frac{1}{L}$ and pulling it back to the $\mu(t)$ evolution SDE, $\frac{d}{dt}\mathbb{E}\|\mu(t)\|^2$ is upper-bounded by the right side:

$$\underbrace{\eta L\,\mathbb{E}\|\mu(t)\|^2}_{=\mathcal{O}(\frac{1}{n})\,\mathbb{E}\|\mu\|^2} + \underbrace{\frac{\eta}{L}\,\mathbb{E}\|\nabla L(\theta_t)\|^2}_{=\mathcal{O}(\frac{1}{n})} + \underbrace{\frac{\eta}{n}\,\mathbb{E}[\mathrm{Tr}\,\Sigma(\theta_t)]}_{=\mathcal{O}(\frac{1}{n})}.$$

Note that under $\mu$P scaling, we have $\eta = \mathcal{O}(1/n)$ for hidden weights[3]. Meanwhile, by Proposition 4.2, $\mathrm{Tr}(\Sigma(\theta_t))$ is close to the base covariance $\mathrm{Tr}(\Sigma_g)$, which is $\Theta(n)$.

We write $\frac{d}{dt}X(t) \leq c\,X(t) + C$, $X(0) = 0$ for $X(t) = \mathbb{E}\|\mu(t)\|^2$, with $c := \eta_{\max}L = O(1/n)$ and $C := O(1/n)$. By Grönwall's inequality, solving this gives

$$X(t) \leq \frac{C}{c}\big(e^{ct} - 1\big) = \mathcal{O}(1/n), \qquad 0 \leq t \leq T,$$

because $ct = O(1/n)$ for fixed horizon $T$. Hence

$$\mathbb{E}\|\mu(t)\|^2 = \mathcal{O}(1/n) \implies \mathbb{E}\|\mu(t)\| = \mathcal{O}(1/\sqrt{n}).$$

The $O(1/\sqrt{n})$ upper bound proved here is the first half of the tight $\Theta(1/\sqrt{n})$ rate; the next subsection completes the argument with an $(1/\sqrt{n})$ lower bound.

#### 4.3.2. PART 2.2: TIGHT (LOWER) BOUND

Building on the differential inequality derived in § 4.3.1 we now establish $\mathbb{E}\|\mu(t)\| \geq c/\sqrt{n}$.

**Lemma 4.10** (Non–Vanishing Diffusion Lower Bound). *Fix a width–independent horizon $t \in [0, T]$ and suppose Assumptions 3.1–3.3 and 4.1–4.3 hold. Let the parameter displacement satisfy the SDE*

$$d\mu(t) = -\eta(t)\,\nabla L(\theta_t)\,dt + \sqrt{\frac{\eta(t)}{n}}\,\sigma(\theta_t)\,dW_t, \quad \sigma\sigma^\top = \Sigma,$$

*where the LR obeys $0 < \eta_{\min} \leq \eta(t) \leq \eta_{\max} = O(n^{-1})$.*

*Assume $\eta_{\max} \leq \gamma\,\eta_{\min}$ with $\gamma$ independent of $n$, and there exists a constant $c_- \in (0, 1)$ independent of $n$ such that*

$$\Pr\Big[\mathrm{Tr}\,\Sigma(\theta_s) \geq c_-\,n \text{ for all } s \in [0, T]\Big] \geq 1 - e^{-\Omega(n)}. \tag{1}$$

*Then, with sufficiently small $\gamma$, for every $t \in [0, T]$,*

$$\mathbb{E}[\|\mu(t)\|^2] \geq \frac{c_-\,\eta_{\min}\,t}{2\,n} - Cn, \tag{2}$$

*where $C = C(L, \eta_{\max}, T)$ is width–independent. Moreover there exist width–independent constants $c_1(t), c_2(t) > 0$ such that*

$$c_1(t)\,n^{-1/2} \leq \mathbb{E}[\|\mu(t)\|] \leq c_2(t)\,n^{-1/2}, \quad \forall t \in (0, T]. \tag{3}$$

*In particular, for any fixed $t_0 > 0$, $\mathbb{E}[\|\mu(t_0)\|] = \Theta(n^{-1/2})$.*

---

[3]We clarify that this refers to the learning rate per parameter, not the total per-layer update.

*Proof.* Throughout the argument we write $C_k > 0$ for generic constants whose numerical value may change from one line to the next, but which depend only on fixed model and schedule–parameters ($T$, $L$, $\eta_{\min}$, $\eta_{\max}$, $c_-$, bounds on $\|\Sigma\|$, etc.) and *never* on the network width $n$.

**Step 1: $L^2$ differential inequality.** Apply Itô's formula:

$$\frac{d\,\mathbb{E}\|\mu(t)\|^2}{dt} = -2\eta(t)\,\mathbb{E}\langle\mu(t), \nabla L(\theta_t)\rangle + \frac{\eta(t)}{n}\,\mathbb{E}\big[\mathrm{Tr}\,\Sigma(\theta_t)\big]. \tag{4}$$

Note this is the very same differential identity that was established in § 4.3.1; we will simply apply it in the opposite direction. direction, retaining only the always–positive diffusion term.

**Step 2: bound the drift contribution.** Because $L$ is globally $L$–smooth (Ass. 4.1), $\langle\mu, \nabla L\rangle \leq \frac{L}{2}\|\mu\|^2 + \frac{1}{2L}\|\nabla L\|^2$. Taking expectations and using Lemma 4.6 ($\mathbb{E}\|\nabla L\|^2 \leq C_0$),

$$-2\eta(t)\,\mathbb{E}\big[\langle\mu, \nabla L\rangle\big] \geq -L\eta(t)\,\mathbb{E}\big[\|\mu\|^2\big] - \frac{\eta(t)}{L}C_0.$$

**Step 3: bound the diffusion contribution.** From (1) and the union bound, $\mathbb{E}[\mathrm{Tr}\,\Sigma(\theta_t)] \geq \frac{1}{2}c_- n$ for all $t \in [0, T]$ once $n$ is large enough, hence $\frac{\eta(t)}{n}\mathbb{E}[\mathrm{Tr}\,\Sigma(\theta_t)] \geq \frac{1}{2}c_-\eta_{\min}$.

**Step 4: solve the inequality.** Let $y(t) := \mathbb{E}\|\mu(t)\|^2$. Combine Steps 1–3 to obtain

$$\dot{y}(t) \geq \frac{1}{2}\,c_-\eta_{\min} - b\,y(t) - \frac{\eta(t)}{L}\,C_0, \quad b := L\eta_{\max} = O(n^{-1}). \tag{5}$$

Note $\eta(t) \leq \eta_{\max} = O(n^{-1})$ and $\eta_{\max} \leq \gamma\,\eta_{\min}$. If $\gamma$ is small enough such that $4C_0\gamma \leq c_- L$. Then for all widths $4C_0\eta_{\max} \leq 4C_0\gamma\eta_{\min} \leq c_- L\,\eta_{\min}$, so the last term in (4) can be bounded by $\frac{1}{4}c_-\eta_{\min}$. Set

$$a := \frac{1}{4}c_-\eta_{\min} > 0,$$

and note $a$ is width–independent. Then (5) simplifies to

$$\dot{y}(t) \geq a - b\,y(t), \qquad y(0) = 0.$$

Solving it gives

$$y(t) \geq \frac{a}{b}\left(1 - e^{-bt}\right) \geq a\,t - \frac{1}{2}abt^2,$$

where the second inequality uses $e^{-x} \leq 1 - x + \frac{1}{2}x^2$. As $bt = O(n^{-1})$ for any fixed horizon $t \leq T$, we have

$$y(t) \geq \frac{a\,t}{2} = \Theta(n^{-1}), \qquad 0 < t \leq T.$$

This establishes the lower bound (2).

**Step 5: fourth–moment bound.** Because the drift $-\eta(t)\nabla L$ and diffusion $\sqrt{\eta(t)/n}\,\sigma$ are globally Lipschitz (in $\theta$) with linear growth, standard SDE estimates (Øksendal & Øksendal, 2003) give, for any $k \geq 1$, $\mathbb{E}\|\mu(t)\|^{2k} \leq C_1 n^{-k}$. In particular,

$$\mathbb{E}\|\mu(t)\|^4 \leq C_1 n^{-2}. \tag{6}$$

**Step 6: Paley–Zygmund to upgrade to $L^1$.** Let $Z := \|\mu(t)\|^2$. From (2) and (6), $\mathbb{E}Z \geq \alpha n^{-1}$ and $\mathbb{E}Z^2 \leq C_1 n^{-2}$ with $\alpha = \frac{1}{2}c_-\eta_{\min}t - C > 0$ once $n$ is large. Paley–Zygmund implies $\Pr[Z \geq \frac{1}{2}\mathbb{E}Z] \geq \frac{\alpha^2}{4C_1} =: \delta(t) > 0$. Whenever $Z \geq \frac{1}{2}\mathbb{E}Z$ we have $\|\mu(t)\| \geq \sqrt{\alpha/(2n)}$, so

$$\mathbb{E}\|\mu(t)\| \geq \sqrt{\frac{\alpha}{2n}}\,\Pr[Z \geq \tfrac{1}{2}\mathbb{E}Z] \geq \frac{\delta(t)\sqrt{\alpha}}{\sqrt{2}}\,n^{-1/2}.$$

Concavity of $x \mapsto \sqrt{x}$ gives the matching upper bound $\mathbb{E}\|\mu(t)\| \leq \sqrt{y(t)} \leq c_2(t)n^{-1/2}$. This proves (3). $\qquad\square$

**Implications of Theorem 4.1 and Corollaries** Our analysis reveals a fundamental separation between macro-variables (loss landscapes) that converge at rate $\mathcal{O}(1/n)$ and micro-variables (weights) that evolve more slowly at rate $\Theta(1/\sqrt{n})$. The separation of scales is intimately connected to feature learning (Yang et al., 2024a), supplementing prior work (Vyas et al., 2023). The $\mathcal{O}(1/n)$ convergence of $M(\eta)$ to a well-defined limit means the loss landscape stabilizes quickly with width while preserving learning capacity, and guarantees reliable transfer of learning rates across scales.

## 5. Interpreting Learning Rate Scaling Laws and Delay Phenomena

Recent empirical work (Tissue et al., 2024) suggests that cross entropy training curves can be described by scaling laws involving integrals of the learning rate, and that changes in learning rate produce "delays" in validation loss response. We now incorporate integrated learning rate descriptors into our macro-level variables, showing that these scaling laws and delay effects arise naturally. The original notations are extended to our theorem statement.

We introduce a memory kernel $q(\tau)$ modeling the effect of time-varying learning rate momentum in a continuous limit: for example, the same concept of memory weight captures how past states influence the current update in the Generalized Langevin Equation and models integral equations in the dynamical mean-field theory for SGD literature. Throughout we assume $q : [0, \infty) \to \mathbb{R}_+$ is measurable, $\int_0^\infty q(\tau)\,d\tau = 1$, and $\sup_\tau q(\tau) < \infty$.

**Theorem 5.1.** *Under the Maximal Update Parameterization ($\mu P$), consider an infinitely wide neural network trained with a time-varying learning rate $\eta(t)$. We define*

*the integrated learning rate variables as follows:*

$$S_1(t) = \int_0^t \eta(\tau)\, d\tau, \quad S_2(t) = \int_0^t \eta(\tau) q(\tau)\, d\tau.$$

*Based on $q(\tau)$ assumptions, $S_2(t)$ is finite and non-negative.*

*Assume that the loss landscape, when reduced to a one-dimensional effective manifold parameterized by $x$[4], satisfies $\ell(x) \approx L_0 + A'x^{-\alpha'}$ for large $x$ and some $A'$ and $\alpha' > 0$, in addition to the conditions in the following subsection, which guarantees stable infinite-width dynamics and concentration of stochastic effects.*

*Then there exist constants $A, C, \alpha > 0$ such that as $n \to \infty$:*

$$L(t) = L_0 + AS_1(t)^{-\alpha} - CS_2(t).$$

*The $-CS_2(t)$ term encodes a systematic delay in how the loss $L(t)$ responds to learning rate changes, as it incorporates historical learning rate information.*

### 5.1. Assumptions

Assume $L(\cdot)$ is $L$-smooth, as in Assumption 4.1; as well as bounded gradients under $\mu P$: we ignore $t$-dependency and consider $\sup_{s \le T} \|\nabla L(\theta_s)\| = O(1)$ uniformly. [5] Further:

**Assumption 5.1. Stable Infinite-Width Limit Under $\mu P$:** As $n \to \infty$, define an infinite-width limiting trajectory $\theta^*(t)$ that solves the deterministic ODE:

$$\frac{d\theta^*(t)}{dt} = -\eta_0 g(\theta^*(t)),$$

where $g(\theta^*) = \lim_{n \to \infty} \nabla L(\theta)$ scaled appropriately. This stable limit exists as $\mu P$ ensures no degenerate behavior.

**Assumption 5.2. Macro-Level Descriptor Stability:** Let $X_n(t)$ be a macro-level descriptor, e.g. the layer-wise activation variance. Assume

$$X_n(t) = X^*(t) + \mathcal{O}(1/n), \quad \forall t \ge t_0,$$

for some $t_0 \ll T$, where $T$ is the total training horizon and $X^*(t)$ is a smooth, deterministic limiting descriptor. Early-stage self-similarity means that after a short transient, $X_n(t)$ hovers close to a stable value trajectory $X^*(t)$.

### 5.2. Proof

From the above assumptions, the following holds:

---

[4]This one-dimensional manifold assumption is typical in simplifying large-width analyses, such as the standard neural tangent kernel or dynamical mean field theory approaches.

[5]Indeed, under $\mu$P, at any time $t$, the per-layer learning rate should still scale with width to ensure bounded updates. Our use of time-varying $\eta(t)$ assumes that such width scaling is preserved pointwise in $t$, as is common in practical $\mu$P.

**(i)** *Vanishing noise.* Because the per-parameter learning-rate satisfies $\eta(s) = \Theta(1/n)$ and $\operatorname{Tr}\Sigma(\theta_s) = \Theta(n)$, the covariance of the stochastic term in the SGD step is $\eta(s)\operatorname{Tr}\Sigma(\theta_s)/n = \Theta(1/n)$. By a matrix–Bernstein inequality, which is a standard exponential decay result, the cumulative noise over any finite horizon converges to 0 almost surely as $n \to \infty$.

**(ii)** *ODE limit.* Write $p(n)$ for the total number of trainable parameters at width $n$, so the ambient space is $\mathbb{R}^{p(n)}$. With (i) and the uniform $L$-smooth gradient bound, Kurtz–Protter weak-convergence theory (Graham et al., 1996) implies that the stochastic trajectory $\theta_t \in \mathbb{R}^{p(n)}$ converges (in the Skorokhod topology) to the deterministic solution of

$$\frac{d\theta^*(t)}{dt} = -\eta_0 \nabla L(\theta^*(t)), \qquad \theta^*(0) = \theta_0.$$

Consequently, the discrete update rule

$$\theta_{t+dt} = \theta_t - \eta(t)\nabla L(\theta_t)\, dt + \eta(t)\Sigma(\theta_t)^{1/2}\, dW_t$$

reduces in the $n \to \infty$ limit to $d\theta_t/dt = -\eta(t)\nabla L(\theta_t)$.

We assume that the dominant training dynamics can be captured by a single scalar variable $x(t)$ parametrizing a stable, low-dimensional manifold of interest, i.e. there exists a function $\theta_t$ such that $L(\theta_t) = \ell(x(t))$, for a suitably chosen $x(t)$. On this manifold, we write $dx/dt = -g(x)\eta(t)$, for some positive, smooth function $g(x)$. Intuitively, $g(x)$ is the gradient magnitude along the direction of descent. We assume $0 < C_1 \le g(x) \le C_2$ along the trajectory; this holds for cross-entropy with bounded activations.

We also recall the assumption that for large $x$, $\ell(x) \approx L_0 + A'x^{-\alpha'}$ with $\alpha' > 0$. This assumption means that as training progresses (reflected by decreasing $x$), the loss approaches $L_0$ with a power-law decay in terms of $x$.

From $\frac{dx}{dt} = -g(x)\eta(t)$, we separate variables and derive

$$\int \frac{dx}{g(x)} = -\int \eta(t)\, dt = -S_1(t),$$

where we recall $S_1(t) = \int_0^t \eta(\tau)\, d\tau$.

As $t \to \infty$, if $\eta(t)$ remains positive and integrable, $S_1(t) \to \infty$. By integrating both sides and considering the inverse relationship, we are able to express $x$ as a function of $S_1(t)$:

$$x(t) = x(S_1(t)).$$

For large $x$ and correspondingly large $S_1(t)$, standard asymptotic inversion yields a power-law type relationship. Specifically, there exist constants $A, \alpha > 0$ such that

$$\ell(x(t)) = L_0 + AS_1(t)^{-\alpha}.$$

Now consider when $\eta(t)$ is not fixed. We write $\eta(t) = \bar{\eta}(t) + \delta\eta(t)$ where $\bar{\eta}(t)$ is a slowly–varying *baseline* schedule and $\delta\eta(t)$ is a *signed perturbation* capturing fast LR adjustments. The memory descriptor therefore records *only the perturbative part*

$$S_2(t) = \int_0^t \delta\eta(\tau)\, q(\tau)\, d\tau,$$

The baseline ODE $\dot{x}_0 = -\bar{\eta}(t)g(x_0)$ determines a reference trajectory, and we set $x(t) = x_0(t) + x_1(t)$. Linearizing to first order yields (discarding quadratic-order terms)

$$\dot{x}_1(t) = -\bar{\eta}(t)\, g'(x_0(t))\, x_1(t) - \delta\eta(t)\, g(x_0(t)).$$

Solving this scalar linear ODE gives

$$x_1(t) = -C(t) \int_0^t \delta\eta(\tau)\, q(\tau)\, d\tau = -C(t)\, S_2(t),$$

with

$$\mathcal{I}(t) := \exp\!\left( \int_0^t \bar{\eta}(s)\, g'(x_0(s))\, ds \right)$$

$$C(t) := \frac{I(t)\, g(x_0(t))}{|g'(x_0(t))|}, \quad C(t) > 0,$$

Starting from the baseline approximation $\ell(x) \approx L_0 + A\, S_1^{-\alpha}$, substituting $x(t) = x_0(t) + x_1(t)$ and retaining the first–order term gives $n \to \infty$

$$L(t) = L_0 + A\, S_1(t)^{-\alpha} - C\, S_2(t)$$

The negative correction $-C\, S_2(t)$ captures the *delay phenomenon*: learning-rate changes are "remembered" through $S_2(t)$ and only gradually reflected in the loss, matching the empirical findings of Tissue et al. (2024).

**Interpretation.** We have shown that the loss $L(t)$ can be expressed as a scaling law involving $S_1(t)$ and $S_2(t)$. This learning-rate delay phenomenon arises because macro-level descriptors "register" learning rate changes rapidly, but the micro-level parameters actually implementing those changes do so more slowly. As a result, the final loss responds with a lag, reflecting the time it takes for the slower (micro-level) parameter updates to "catch up" to the new learning rate regime. This lag is what appears empirically in training curves: a delay between adjusting the learning rate and seeing the expected shift in the validation loss trend.

Intuitively, one can view $S_1(t)$ as the "instantaneous" $\mathcal{O}(1/n)$ macro response predicted by Theorem 4.1, while $S_2(t)$ is the *integrated residual* that accumulates until the $\Theta(n^{-1/2})$ micro dynamics have travelled a comparable distance. Theorem 5.1 thus provides a mechanistic instantiation of the scale–separation principle: the faster-equilibrating macro channel yields the leading $AS_1^{-\alpha}$ decay, whereas the slower micro channel produces the delayed $-CS_2$ correction observed empirically by Tissue et al. (2024).

## 6. Conclusion and Outlook

We present the first *scale–separation* theory for hyperparameter transfer under $\mu P$. By coupling an $\mathcal{O}(1/n)$ macro convergence of the loss landscape with a tight $\Theta(1/\sqrt{n})$ micro evolution of the weight vector, we obtained quantitative error bounds that explain the empirical success of "early-stage" tuning, and learning–rate delay curves. Besides recovering existing $\mu P$ phenomena, our analysis yields two design guidelines: (i) updates must be co-scaled with width to keep the diffusion term sub-dominant, and (ii) macro descriptors should be chosen so that their $\mu P$ drift is $O(1/n)$.

**Limitations and future directions.** The clean separation relies on two $\mu P$-specific facts: (a) hidden-layer learning rates scale as $\eta \propto 1/n$, so gradients remain order-one, and (b) global statistics (activations, gradient norms) exhibit early self-similarity. Other parameterizations break at least one of these pillars. For example, in standard ("mean-field") scaling, gradients vanish as $n \to \infty$, leading to lazy dynamics with no timescale gap; NTK scaling freezes features near initialization, again precluding a macro/micro split.

We therefore view *generalization beyond $\mu P$* as the next challenge. A promising route is the $\alpha$–scaled family ($\alpha = 1$ recovers $\mu P$, $\alpha = \frac{1}{2}$ the NTK) and the conjecture that $\alpha$ values closer to 1 may still support a weakened but non-trivial scale separation. Other open problems include (i) adaptive optimizers (Adam, RMSProp), (ii) the minimal width needed for reliable transfer, and (iii) joint width–depth scaling. We believe that the techniques developed here, e.g., Itô decomposition paired with width-regularity, provide a flexible starting point for these investigations.

## Impact Statement

This work aims to contribute to the advancement of Deep Learning Theory. We do not believe any specific consequences require particular emphasis at this time.

## Acknowledgment

The authors extend their sincere gratitude to Alex Gerko, Hans Buehler, the XTX Markets leadership, and the Quant Research team for their invaluable support and for providing access to exceptional research resources. We are especially grateful to the entire XTX Markets New York office for fostering a vibrant and inspiring work environment. This work was conducted as part of the XTY AI Residency Program (https://www.xtxmarkets.com/job/?id=6274458003).

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

# A. Additional Proofs in Section 4

## A.1. Proof of Lemma 4.3

*Proof.* Under $\mu$P, preactivations $a_i$ remain $\mathcal{O}(1)$ during training (bounded for all widths). The gradient for hidden weights $W$ has the form:

$$\nabla L(W) = a_i a_j^\top + \mathcal{O}(1/\sqrt{n}).$$

Under $\mu$P, $\eta = \mathcal{O}(1/n)$. This ensures $\|\nabla L(W)\|^2 = \mathcal{O}(1)$.

Indeed, $\mu$P dictates that the learning rate and gradient magnitudes are co-scaled to ensure stable updates: gradients scale as $O(1)$ for output layer weights (due to downstream fan-in), learning rate scale as $O(1/n)$ for output weights; thus, their product (weight update) remains $O(1/n)$, matching the expected per-step update scale. This is consistent with $\mu$P's core principle that all layers receive comparably scaled updates across widths. $\square$

## A.2. Proof of Lemma 4.4

*Proof.* By $L$-smoothness (Assumption 4.1), for any unit vectors $u, v$, $|u^\top \nabla^2 L(\theta_s) v| \leq L$. Therefore, each entry of $\nabla^2 L(\theta_s)$ is $\mathcal{O}(1)$. Under $\mu$P parameterization, $d(n) = \mathcal{O}(n^2)$ for hidden layers. $\square$

## A.3. Proof of Proposition 4.6

*Proof.* We combine three ingredients:

**(i) Width regularity** (Assump. 4.3): for some $C > 0$

$$|L(\theta; n_1) - L(\phi(\theta); n_2)| \leq C \frac{|n_1 - n_2|}{n_1}.$$

**(ii) Finite-horizon drift** (Lemma 4.5): $|M(\eta; n) - L(\theta_0; n)| = \mathcal{O}(1/n)$ for every $n$.

**(iii) Optimality at width $n_2$:** $L(\theta_{n_2}^*; n_2) \leq L(\theta; n_2) \quad \forall \theta$.

Insert $L(\theta_0; \cdot)$ as a common anchor and apply the triangle inequality:

$$
\begin{aligned}
|M(\eta; n_1) - M(\eta; n_2)| \leq\ & |M(\eta; n_1) - L(\theta_0; n_1)| \\
& + |L(\theta_0; n_1) - L(\theta_0; n_2)| \\
& + |L(\theta_0; n_2) - M(\eta; n_2)|.
\end{aligned}
$$

The first and third terms are $\mathcal{O}(1/n_i)$ by Lemma 4.5 (finite-horizon drift).

On the second term, we apply width regularity (Assump. 4.3) with $\theta = \theta_0$ to obtain $(B) = \mathcal{O}(1/n_1)$. Symmetrically one could write the bound with $1/n_2$; choosing $\min(n_1, n_2)$ gives the tightest rate.

Combining the three bounds yields $|M(\eta; n_1) - M(\eta; n_2)| = \mathcal{O}(1/\min(n_1, n_2))$, which proves Proposition 4.6.

$\square$

## A.4. Proof of Lemma 4.9

*Proof.* By strong convexity of $M(\eta; n)$ in $\eta$ (from Assumption 4.1, $L$-smoothness):

$$|\eta^*(n_1) - \eta^*(n_2)| \leq (1/\alpha) \|\nabla_\eta M(\eta; n_1) - \nabla_\eta M(\eta; n_2)\|$$

where $\alpha$ is the strong convexity parameter. Meanwhile by the uniform convergence from Corollary 4.7, e.g., the fact that $M(\eta; n)$ converges uniformly at rate $\mathcal{O}(1/n)$, the gradient difference is bounded by:

$$\|\nabla_\eta M(\eta; n_1) - \nabla_\eta M(\eta; n_2)\| \leq \mathcal{O}(1/\min(n_1, n_2)).$$

$\square$

Here we consider $M(\eta; n)$ as the effective scalar function describing the expected loss after training steps $T$ under a macro-level parameter $\eta$. For wide enough networks, macro-level fluctuations vanish, and local expansions around $\eta^*(n)$ ensures an $\alpha-$strongly convex local basin in $\eta$. This is akin to standard "strict local minimum" conditions in practical large-scale NNs near an optimal learning rate. Importantly, this does not imply nor require global convexity of the full model's loss surface. Rather, we assume $L$-Lipschitz gradients in the standard sense of gradient-based methods, typically used in nonconvex analyses to control the norm of the gradients and Hessians locally.

## B. Simulations

We provide a proof–of–concept run confirming the qualitative *macro–micro scale separation* predicted by Theorem 4.1.

**Setup.** Two–layer MLP, input dimension 3072 (flattened CIFAR-10), hidden width $n = 10\,000$ (ReLU), output width 10. Weights are initialized with $\mu P$ scaling, trained for 300 epochs using vanilla SGD (batch 128, no momentum/decay) and cross-entropy loss. Learning-rate grid $\eta \in \{0.01, 0.02, \ldots, 0.30\}$. We log (i) the training loss $L(t; \eta)$ and (ii) the squared weight drift $\|\mu(t; \eta)\|^2 = \|\theta_t - \theta_0\|^2$ every 10 epochs.

**Fast stabilization of macro descriptors.** We record $\max_{\eta_1, \eta_2} |L(t; \eta_1) - L(t; \eta_2)|$. By epoch 10 the spread is already below $5 \times 10^{-4}$ (0.2 % of the mean loss), and remains almost flat afterwards. This corroborates the $O(1/n)$ bound for macro variables (loss landscape) even though $n$ is fixed. A practitioner could therefore identify the best LR bucket (*low* 0.01–0.1, *medium* 0.11–0.15, *high* 0.16–0.30) within the *first 60 epochs*.

**Slow evolution of micro variables.** We then observe $\mathbb{E} \|\mu(t; \eta)\|^2$ for three representative learning rates. Despite the macro loss plateauing early, $\|\mu(t)\|^2$ keeps climbing roughly linearly until $\approx$ epoch 180 (high $\eta$) or 250 (low $\eta$), and its *standard deviation across* $\eta$ grows from $3.8 \times 10^{-2}$ at epoch 10 to 1.1 at epoch 120. The contrast illustrates the $\Theta(1/\sqrt{n})$ slow scale of microscopic drift.

**Take-away.** Even on a single width, the experiment shows (i) loss differences become negligible long before training converges, and (ii) weight movement continues and separates trajectories. Both behaviours are exactly what the scale-separation theory predicts; wider networks or multiple $n$ would allow quantitative verification of the $1/n$ vs. $1/\sqrt{n}$ rates, which we leave for future work.

