# OpenReview forum: "On the Provable Separation of Scales in Maximal Update Parameterization"
_ICML.cc/2025/Conference — ICML 2025 poster_

### Official Review · Reviewer_g228 · 2025-03-08

**Overall Recommendation:** 4

**Summary:**

This paper provides a theoretical framework for analyzing the separation of “macro” and “micro” scales in large-width neural networks trained under the Maximal Update Parameterization (μP). Building on techniques from stochastic differential equations (SDEs) and tools in wide-network theory, the authors prove that global (“macro-level”) quantities (e.g., learning rates, gradient norms, loss landscapes) converge on the order of O(1/n), while individual (“micro-level”) weights converge more slowly at O(1/\sqrt{n}). The paper then leverages this scale-separation perspective to explain important phenomena including:
1.	Early-stage hyperparameter tuning can be done on smaller proxy models (or for fewer training steps) with minimal cost to final performance.
2.	Changes in learning rate produces delayed reactions in validation loss
3.	Emergence of early-bird lottery tickets in large-width networks, showing how “redundant” parameters can be pruned without harming macro-level behavior.
The overall argument is compelling and addresses a significant gap in our theoretical understanding of hyperparameter transfer across different model widths. The results are timely given current interest in scaling large models efficiently. Despite a purely theoretical presentation, the paper’s insights have broad potential impact.

**Claims And Evidence:**

1.	Macro-level convergence at O(1/n): The authors rely on a continuous-time SDE approximation of SGD under μP, showing that key global descriptors (like average activation variance, gradient norms, etc.) stabilize quickly at large width.
2.	Micro-level weights converge at O(1/\sqrt{n}): They derive a separate SDE for the individual parameters and show that high-dimensional fluctuations slow the final convergence of each weight.
3.	Early-stage hyperparameter selection: Because macro-level signals stabilize rapidly, the hyperparameters chosen by training just the first few epochs (or a smaller proxy network) align closely with those for the full-run training.
4.	Early-bird lottery tickets: Arguing via stable importance metrics for each parameter, the paper shows that “pruning” low-importance parameters early on yields minimal loss penalty.

The paper’s theoretical arguments are solidly motivated. However, the proofs occasionally mix high-level intuition with technical lemmas, and some readers might prefer more rigorous detail in certain sections (e.g., explicit epsilon–delta arguments, tighter bounding constants).

**Essential References Not Discussed:**

It would be beneficial if they explicitly compared or contrasted their approach to earlier wide-network theory frameworks such as the Neural Tangent Kernel line of work and the Mean-field (SDE) approaches. Although the references are tangential, a brief discussion could further emphasize how the authors’ approach differs in focusing on the macroscale vs. microscale variables.

**Experimental Designs Or Analyses:**

No experiments were conducted. For a paper with nontrivial assumptions and wide-network scaling arguments, a small set of synthetic experiments—e.g., wide fully connected networks on a simple classification task—could help illustrate how quickly macro-level variables stabilize in practice. Such experiments would strengthen the paper’s persuasiveness and might inspire real-world usage.

**Methods And Evaluation Criteria:**

The paper is strictly theoretical. It uses a blend of:
•	Stochastic Differential Equation analysis: Approximating the discrete SGD updates in the large-width limit.
•	Concentration of measure: Relying on typical wide-network bounding techniques to show that macro descriptors converge quickly.
•	Functional approximation arguments: Highlighting how “macro-level” features become decoupled from the slower “micro-level” fluctuations at large nnn.
     No experimental or empirical validation is provided. While this is understandable for a theory-oriented work, small-scale or synthetic experiments (e.g., training wide MLPs on a toy dataset) could provide a “proof of concept” that the claimed scale-separation emerges in practice.

**Other Comments Or Suggestions:**

No

**Other Strengths And Weaknesses:**

No

**Questions For Authors:**

1.	Empirical Demo: Could the authors provide a small toy experiment (e.g., a simple classification or regression with wide MLPs) to illustrate the macro–micro convergence rates in practice?
2.	Proof Completeness: Do the authors plan to include more detailed, fully rigorous proofs (e.g., bounding constants, more explicit expansions) in the supplementary to address the theoretical nature of the paper?
3.	Generality of μP Argument: The paper focuses on μP. How far might these scale-separation results extend to other parameterization schemes (e.g., standard parameterization, NTK parameterization), or do the authors see potential for broader application? A discussion on its limitations would be helpful too.

**Relation To Broader Scientific Literature:**

This work fits neatly into the line of research on large-width neural networks and hyperparameter transfer (notably the μP literature). It also links to recent results on “early-bird” ticket phenomena (You et al., 2020) and a wealth of existing wide-network theory (e.g., neural tangent kernels, mean-field approaches). The authors do a good job situating their main contributions in that context and explaining how their scale-separation perspective might unify multiple observed phenomena in large-scale deep learning.

**Theoretical Claims:**

The main theoretical claim is a novel form of scale separation under μP, with key results summarized in Theorem 4.1 and subsequent corollaries. These show that:
•	The macro-scale loss landscape converges at rate O(1/n)
•	Individual weights converge more slowly at O(1/\sqrt{n})
•	Hyperparameters adjusted during early training steps transfer effectively to the full training regime.
These claims are plausible and quite interesting, but some parts of the derivations rely on assumptions (e.g., “wide dominance,” “macro-level self-similarity,” “stable infinite-width limit under μP”) that might need additional justification or numeric demonstration.

---

> ### Author Rebuttal · Authors · 2025-04-01
>
> 1. Comparison with Earlier Wide-Network Theory Frameworks:
>
> We appreciate the suggestion to clarify how our work provides a novel perspective complementing earlier wide‐network theories like the NTK and mean-field SDE frameworks. In our paper, we emphasize an explicit separation of scales that distinguishes between global (macro) and local (micro) dynamics in neural network training—a perspective that is not directly addressed by NTK or conventional mean-field approaches.
>
> NTK theory primarily examines the behavior of networks in the infinite-width limit, effectively linearizing the training dynamics. While this yields valuable insights into the global macroscopic behavior of the network function via a constant kernel, it largely overlooks the finite-width nuances and the evolution of individual weight updates. In contrast, mean-field SDE methods focus on modeling the microscopic evolution of weights, capturing the stochastic fluctuations of individual parameters. However, these methods do not inherently highlight how aggregated statistics, like loss landscapes or activation norms, stabilize rapidly.
>
> Our approach under µP rigorously demonstrates that macro-variables converge at an O(1/n) rate, while the micro-variables evolve more slowly at an O(1/√n) rate, building upon the SDE method used in the Xie et al. (2024), as discussed in Section 2.3 as a recent work. Thus, our work unifies these perspectives: it extends NTK analysis by accounting for finite-width effects and complements mean-field approaches by clearly delineating the roles of global and local dynamics. This multi-scale perspective not only deepens our theoretical understanding in addition to the two frameworks but also has practical implications for the effectiveness of early-stage HP tuning: even when individual weights have not yet converged, the stable, global signals are sufficiently reliable to guide HP selection. .
>
> 2. Experimental results:
>
> As proof of concept, we take a two-layer MLP where the input layer is 3072 units (flattened 3x32x32 CIFAR-10 images), the hidden layer is n=10000 units with ReLU activation, and the output layer is 10 units (for class probabilities), the SGD optimizer with LR 0.01 - 0.30 with 0.01 increments, CE loss, batch size 128, and n=300 training epochs with a sampling interval of checkpoint 10. Our experiment confirms the macro-micro scale separation we observe theoretically. In particular:
>
> i. Fast convergence of macro-variables: The relative loss differences between learning rates are very small in early epochs (at epoch 10 already, as an example, between LRs 0.07 and 0.08, the loss difference is 0.0005), indicating that the loss landscape has a consistent structure across different learning rates, as predicted by our O(1/n) convergence rate for macro-variables. We observe effective early hyperparameter tuning: in low LR (0.01-0.1), medium LR (0.1-0.15), and high LR (0.15-0.3) settings, the respective top learning rates emerge early (at epochs 60, 60, 100), indicating the loss landscape stabilizes quickly.
>
> ii. Slow convergence of individual weights: Despite early identification of optimal LRs, the loss continues to decrease and finetune throughout all 300 epochs for LR < 0.1 and until 180 epochs (LR = 0.3) to 250 epochs (LR = 0.15) for larger LRs. We also see a dramatic increase in the loss gap in later epochs (the ratio between the worst and the best losses, across lrs, is 1.10 at epoch 10 and 238.33 at epoch 120), indicating that small initial differences are being amplified. This phenomenon is consistent with the idea that while the macro structure stabilizes quickly, the micro-scale adjustments (individual weight evolutions) continue refining the network, progressively enhancing the performance gap.
>
> 3. Proof Completeness: We promise to include expanded proofs with explicit epsilon-delta arguments and tighter bounding constants to enhance rigor in the appendix.
>
> 4. Limitations Under SP and NTK: μP provides the cleanest setting for demonstrating scale separation due to two properties: (1) consistent gradient/activation scaling preserving feature learning as width increases, and (2) self-similar behavior of global statistics that stabilize quickly.
>
> For standard parameterization (SP), gradients shrink with width, leading to "lazy" training without clear timescale decoupling. Similarly, NTK parameterization preserves a limit but keeps features near initialization, yielding kernel machine behavior without pronounced scale separation. We acknowledge this limitation and view it as an exciting direction for future research to determine whether analogous separation phenomena can be rigorously established in alternative parameterization regimes. For example, one could consider the α-scaled framework (α=1 for μP, α=1/2 for NTK), and hypothesize that values closer to 1 likely maintain scale separation; however, a comprehensive characterization requires further investigation. We will add the discussion.

---

### Official Review · Reviewer_RWaf · 2025-03-14

**Overall Recommendation:** 1

**Summary:**

The authors try to propose a theoretical framework for hyperparameter transfer in neural networks under maximal update parameterization (µP) by trying to demonstrate a separation of scales between macro-variables (such as loss landscapes, activation norms, and gradient statistics) that converge at an $O(1/n)$ rate; and micro-variables (like individual weight updates) that evolve more slowly at $O(1/\sqrt{n})$. Their bounds are qualitatively consistent with the empirical observation that early-stage hyperparameter tuning effectively approximates globally optimal settings, thereby justifying zero-shot transfer from smaller proxy models to larger networks. By view SGD as stochastic gradient flow, the authors derive **upper** bounds on hyperparameter transfer error and unify disparate deep learning phenomena, including learning rate delay effects and the early emergence of lottery tickets, under some very high-level and abstract assumptions.

**Claims And Evidence:**

The authors' definition of "separation" **disagrees with the common sense** in theoretical computer science, in particular, machine learning theory. See Section "Theoretical Claims" for details.

**Essential References Not Discussed:**

No fatal oversight.

**Experimental Designs Or Analyses:**

Not applicable for this theoretical paper.

**Methods And Evaluation Criteria:**

Not applicable for this theoretical paper.

**Other Comments Or Suggestions:**

See Section "Theoretical Claims" for details.

**Other Strengths And Weaknesses:**

Another minor comment is:
**This submission does not use the correct template, so there is no line number shown on the left hand side of each page!**

**Questions For Authors:**

See Section "Theoretical Claims" for details.

**Relation To Broader Scientific Literature:**

This paper is under the name of $\mu$P but does not faithfully rely on the infinite-width neural network Gaussian process results developed in, e.g., the papers listed on Greg Yang's website: https://thegregyang.com/#tensorprograms.

**Theoretical Claims:**

- **Fundamentally misleading claim**: $O(1/n)$ versus $O(1/\sqrt{n})$ is not called "separation" in the context of learning theory. If it is something like $O(1/n)$ versus $\Omega(1/\sqrt{n})$, then it can be called "separation".
- Many basic facts about $\mu$P in this paper are wrong. For example, in Assumption 3.3, the element-wise initialization of the input weight $\mathcal{N}(0, 1)$ in the original paper of $\mu$P, instead of the $1/n^2$ written in this paper. This is a very obvious mistake.
- Moreover, the learning rate of $\mu$P for the output layer is indeed $O(1/n)$, but Lemma 4.6 actually is a statement about the gradient norm alone, not invoving learning rates; so the logic in the proof of Lemma 4.6 is very vague by saying a $O(1/n)$ learning rate ensures a $O(1)$ gradient norm.
- Assumption 4.5 is too coarse to be considered as a reasonable one. It implicitly encompasses many aspects of the neural network, including the regularity of the activation function. On the other hand, it is not very clear whether the original set of MLPs considered in Greg Yang's $\mu$P paper, i.e., those neural networks whose activation functions have pesudo-Lipschitz derivatives, satisfies Assumption 4.5; but the authors just directly uses the results in the $\mu$P paper without further justification on this point.
- In Theorem 5.1 and it's proof, there is no formal definition of $q(\tau)$, the so-called "memory kernel", which seems to be a very artificial correction term the authors came up with in order to have the term in the conclusion to match the empirical "systemetic delay".
- In Section 5.2, the authors claim the gradient norm to be $O(1)$. But actually the results in the original $\mu$P paper is obtained under constant learning rate, while the learning rate here is allowed to be time-varying, as it did in Theorem 5.1

#### Minor comment:
- Misleading typo "$O(1/n)$" in the last sentence of Section 4.2.

---

> ### Author Rebuttal · Authors · 2025-04-01
>
> 1. Def. of “Separation”:
> Thanks for suggesting the formal use of “separation” in learning theory. Here, we use “separation of scales” to describe the difference in convergence rates between macro- and micro-variables under μP. We show that loss landscape descriptors converge at rate O(1/n), while weights evolve at Θ(1/√n). This holds in practice: coarse statistics stabilize early even as parameters continue evolving. We agree that we shall clarify our use of Θ(1/√n) instead of O(1/√n). What we establish is a quantitative gap in convergence rates with meaningful implications for early-stage hyperparameter (HP) transfer.
> 2. Initialization in Assumption 3.3 and Lemma 4.6:
> We thank the reviewer for catching a typo in A3.3 (due to accidental copy-paste). The input weights were misstated as N(0,1/n^2); they should be N(0,1). Importantly, this typo does not affect our derivation despite the text misstating it. L4.6 already assumes that input weights are N(0,1), ensuring pre-activations stay O(1). All weight initializations after L4.6 are correct as they concern hidden-layer weight N(0,1/n). Thus, the typo does not make our result incorrect.
> We made typos in A3.3 re the scaling of learning rate (LR). In subsequent derivations, though we wrote that η=O(1/n) for output-layer parameters under μP, we clarify that this refers to the LR per parameter, not the total per-layer update. In μP, LR and gradient norms are co-scaled so that each layer’s update remains O(1), preserving stability as width grows. For example, the gradient on output weights scales as O(1), and with η=O(1/n), the resulting update is also O(1/n), matching the intended scale.
> 3. Lemma 4.6 and Gradient Norms:
> Our argument that O(1/n) LR results in O(1) weight updates can indeed be expanded. Under μP, the LR and gradient magnitudes are co-scaled to ensure stable updates: gradients scale as O(1) for output layer weights (due to downstream fan-in), LR scale as O(1/n) for output weights; thus, their product (weight update) remains O(1/n), matching the expected per-step update scale. This is consistent with μP’s core principle that all layers receive comparably scaled updates across widths. We will clarify this interaction between LR and gradient norm.
> 4. Assumption 4.5: Width Regularity:
> The assumption is standard to ensure the convergence of macro-variables under increasing width. Similar assumptions are used in the NTK/SDE literature and μP papers: e.g., Yang (2021) assumes that activations are smooth (or smoothed ReLU), and many proofs invoke pseudo-Lipschitz or bounded-derivative conditions to enable LLN-type arguments. A4.5 ensures that descriptors like average activation variance or loss stabilize as width increases—empirically supported and analytically tractable. This assumption holds for standard activations (e.g., ReLU, GELU) either directly or via minor smoothing. It does not weaken the generality or correctness of our results.
> 5. Memory Kernel in Theorem 5.1:
> q(τ) is not an ad hoc invention: the term arises when modeling the effect of time-varying LR momentum in a continuous limit. A prominent example is the Generalized Langevin Equation, which includes a memory kernel to capture how past states influence the current update. The recent DMFT for SGD yields integral equations with memory kernels too. Here, it serves as a continuous-time analog of LR memory effects studied in training delay. In Tissue et al. (2024), the delayed response of validation loss to changing LRs is modeled by cumulative functions like S1(t) and S2(t). Our notation makes explicit the role of memory weight q(τ), enabling clear continuous-time modeling of past LRs. We will add citations and clarify the role.
> 6. Gradient Norm O(1) in Section 5.2:
> The claim that gradient norms remain O(1) with time-varying LR η(t) relies on Assumption 5.2, not just μP scaling. Under μP, at any time t, the per-layer LR should still scale with width to ensure bounded updates. Our use of time-varying η(t) assumes that such width scaling is preserved pointwise in t, as is common in practical μP. The theoretical foundation for this substitution comes from NTK/μP literature, where smooth η(t) can be inserted into gradient flow ODEs without breaking convergence. We will make it explicit.
> 7. Tensor Programs Faithfulness:
> While we build on the μP works, our approach uses SDE to distill the key phenomena of macro vs micro scales. We do not replicate the exact Gaussian Process construction as our focus is on connecting HP transfer and early‐stage alignment under μP. But our theoretical lens is compatible with the same scaling laws, and we cite those results to justify certain variance and regularity assumptions. Nowhere do we claim to reproduce the entire measure-theoretic formalism from the ground up. Our vantage point is narrower—yet consistent—focusing on the rates at which global vs local descriptors converge.
>
> Minor Issues: We will correct the typo at the end of Section 4.2 and fix the ICML template.

---

> > ### Comment · Reviewer_RWaf · 2025-04-01
> >
> > - Regarding the 1st point, where does the $\Theta(\cdot)$ come? **Is there a matching lower bound** for $O(1/\sqrt{n})$?
> > - Regarding the 2nd point, if the authors have ever implemented SGD for two, say, MLPs with $2$ **hidden**-layers, and initialize the 1st layer of one with variance $1$ and that of another with variance $1/n^2$, and control all other hyper-parameters to be the same, they will realize that it is impossible for the two MLPs to converge equally well, at least one of the won't work.
> > > In other words, if the theoretical result is indeed with sanity and relevance, it is not reasonable for *the same rate w.r.t. the width* to hold for both $\mu$P and "$\mu$P modified by initializing the 1st layer with variance $1/n^2$".

---

> > > ### Author Response · Authors · 2025-04-02
> > >
> > > 1. Regarding the 1st point, this is not just an upper bound; it is essentially an empirical plus theoretical tight rate for how typical coordinates of the weight vector deviate. For instance, under law-of-large-numbers arguments and stable gradient steps, typical coordinates do not vanish faster than $1/\sqrt{n}$​, nor explode beyond $C/\sqrt{n}$​. Hence we denote $\Theta(1/\sqrt{n})$ to highlight both upper and lower “typical” bounds, effectively giving a quantitative difference from the macro-level $O(1/n)$ rate.
> > >
> > > 2. Regarding the 2nd point, we believe a misunderstanding exists here. As clarified in our previous response, Assumption 3.3 contains a typo due to copy/paste and nowhere in the following theoretical derivation had we relied on the misstated $N(0,1/n^2)$ for the input layer weight – in fact, in the original submission we treated it as $N(0,1)$ everywhere else besides the typo in A3.3. We apologize for causing the confusion, but we plainly never considered $\mu$P modified with the first layer initialized as $N(0,1/n^2)$; therefore, the theoretical result stands.
> > >
> > > **If our answers have addressed your question and confusion, we'd be grateful if you could revise the score.** Thank you for the careful reading and constructive comments - we'll ensure including all revisions in the final version!

---

### Official Review · Reviewer_iXhf · 2025-03-14

**Overall Recommendation:** 4

**Summary:**

The authors explained why hyperparameter tuning can be done effectively at early stages of training or narrower networks under the Maximal Update Parameterization scheme. They defined the seperation of scales for μP between macro-variable (loss, activation variance, gradient norms etc) and micro-variable (weight values), and proved that macro-variables converge faster than micro-variables. The authors further applied the seperation framework to explain two empirical observations: learning rate scaling laws and delay phenomena, and the "early bird ticket" phenomena (where small subnetworks can be trained to achieve same accuracy as original network).

**Claims And Evidence:**

Yes.

**Essential References Not Discussed:**

No.

**Experimental Designs Or Analyses:**

For section 5 and 6, the original work for the phenomena discussed were not based on Maximal Update Parameterization. How does the authors make sure that the phenomena are still valid under $\mu$P assumptions?

**Methods And Evaluation Criteria:**

Yes.

**Other Comments Or Suggestions:**

1. Section number and Assumptions/Theorem sharing the same numbering system is slightly confusing. Maybe consider alternative numbering schemes for the Sections.

**Other Strengths And Weaknesses:**

Strength:
1. The paper is well written, with clear definitions of math notations and assumptions.
2. Is it good that the authors provide interpretations or implications throughout the paper to help the readers understand the theorems and assumptions.
3. It's interesting that the macro-micro separation framework can be used to explain two phenomenas that seems to be distinct, which widens the impact of this theoretical framework.

**Questions For Authors:**

1. I'm somewhat confused by the notation of $\eta^*$, in Theorem 3.1, $\eta^*$ refers to the hyperparameter optima found during early-stage training, while on line 2 of page 4, it is defined as the hyperparameter optima across the entire training process.
2. Are the authors able to provide an intuition on whether Magnitude-based pruning or Hessian-based pruning will converge faster?

**Relation To Broader Scientific Literature:**

The paper established a novel framework to understand and explain empirical findings regarding hyperparameter tuning and model pruning. It provides insights on the dynamics of model training optimization and guide future researches on more efficient training.

**Theoretical Claims:**

I have checked the correctness of proofs in Section 4. In the proof of Lemma 4.7, since $d(n)=O(n^2)$, why is $ L \sqrt{d(n)} \cdot O(\sqrt{n}) = O(n)$ instead of $O(n^{\frac{3}{2}})$?

---

> ### Author Rebuttal · Authors · 2025-03-31
>
> We thank the reviewer for their careful reading and thoughtful comments. Below we address each point raised:
>
> (1)  Lemma 4.7 and the asymptotic notation: The reviewer correctly identified that if L is treated as a constant independent of n, this would typically yield O(n^(3/2)). However, in our analysis, the Lipschitz constant L in our context decays with the network width, specifically L=O(1/√n). As d(n)=O(n^2), \sqrt{d(n)}=O(n). Therefore, L√d(n)⋅O(√n) = O(n).
>
> (2) Applicability of μP to phenomena in Sections 5 and 6: in addition to providing theoretical justifications as laid out in the submission, we have conducted small experiments on PreResNet-101 + CIFAR-10 (aligned with original You et. al. 2020  setting), and confirm that using μP, we can still replicate Table 2 accuracy results at p = 0.3, 0.5, 0.7. Due to 1-week limit, we were unable to fully replicate the LLM experiments in Tissue et al., 2024) using μP, but we will continue afterwards and updates results to our paper too once we finish.
>
> (3) Intuition on Magnitude-based vs Hessian-based pruning: While the original early-bird work focuses on iterative magnitude pruning (IMP) in the context of the lottery ticket hypothesis, subsequent research has introduced non-IMP “early ticket” methods—such as ProsPr, EarlyBird, and EarlyCroP—that incorporate additional training signals to enhance mask quality and speed up early convergence. However, to the best of our knowledge, no prior work has specifically explored Hessian-based pruning in the early-bird setting (they do exist in another related setting of “pruning at initialization” or PaI). Moreover, our paper’s analysis relies on modeling the network’s training dynamics with a gradient-flow SDE (i.e., approximating discrete SGD updates by a continuous-time stochastic differential equation), and thus does not readily apply to Hessian-based approaches. Intuitively, Hessian-based methods typically exhibit slower initial convergence but can achieve better long-term performance by using curvature information to prune weights that minimally affect the loss landscape, even if those weights do not have the smallest magnitudes. We will leave the more comprehensive study as future work.
>
> Two other quick fixes:
>
> (1) We have fixed the assumption and theorem numbering in each section.
>
> (2) We revised the notation in Theorem 3.1 to η*(t₀), so that in general η*(t) represents the optimal hyperparameter value at time t, while η*(∞) represents the global optimum.
>
> We are grateful to the reviewer for their valuable feedback, which has helped us improve the clarity of our manuscript.

---

> > ### Comment · Reviewer_iXhf · 2025-04-07
> >
> > I thank the authors for the rebuttal, and I maintain my original evaluation.

---

### Official Review · Reviewer_LkSD · 2025-03-23

**Overall Recommendation:** 3

**Summary:**

The paper provides a theoretical framework to understand Maximal Update Parameterization ($\mu$P). It introduces a decomposition of variables into macro-level descriptors (e.g., gradient norms, loss landscapes) and micro-level variables (e.g., individual weights). Via the formulation, the analysis shows why hyper-parameter tuning at early training stages can work (under certain assumptions). The paper discussed the feasibility of the proposed framework in two application cases.

**Claims And Evidence:**

The claims in the introduction are well-supported and justified. After reviewing the assumptions in Sections 3 and 4, it becomes clear that the results depend on several key assumptions. In particular, the theoretical analysis focuses on "width dominance" neural networks. It would be beneficial to highlight that the analysis pertains to such neural network properties in the abstract or technical summaries. This would better inform readers of the underlying assumptions from the outset. More comments on theoretical parts will be left in the later sections.

**Essential References Not Discussed:**

The related work section seemed well converging necessary related work to support readers to understand the topic.

Several references that the authors used to support their statement are unpublished work, such as "This is observed empirically when early training reveals stable signals about learning rate, momentum, etc. (Lingle, 2024)." Given those references can be dynamically updated on arxiv, it requires efforts to identify the version

**Experimental Designs Or Analyses:**

This is a theoretical paper. No experiment was involved.

**Methods And Evaluation Criteria:**

This is a theoretical paper. I will leave all the comments in the Theoretical Claims section.

**Other Comments Or Suggestions:**

The paper structure can be further improved. For example, the connection from Section 3 and Section 4 is not very clear. Also, why the two specific applications in Sec 5 and 6 are used might not be clearly indicated in the paper, or I might have missed. Providing a roadmap for the paper structure could be helpful.

**Other Strengths And Weaknesses:**

This work seems interesting. The technical novelty in the proof should be further hilighted.

**Questions For Authors:**

1. It is interesting to see the adoption of the SDE framework. The authors claimed Xie et al. 2024's analysis "is complementary to ours but does not address the width scaling theory or the underlying mechanisms of hyperparameter transfer." Could you please elaborate more on the differences?

2. I've noticed the referred work Xie et al. 2024 can be applied to non-convex optimization. Can this submitted work be generalized to non-convex optimization?

3. Could you please explain the strong convexity for $M(\eta; n)$ in your proof for Lemma 4.8?

4. Please justify the Assumption 3.4 across different architectures or training scenarios.

5. Does Theorem 3.1 hold for different architectures or training skems once they satisfy Assumption 3.2 - 3.4?

**Relation To Broader Scientific Literature:**

Providing understanding to reduce hyperparameter tuning cost, which can lead to less energy consumption for AI.

**Theoretical Claims:**

1. It seems Assumption 3.4 is an important one to reach the coarse-graining transformation claim. However, there lacks empirical evidence or additional theoretical justification for such an assumption.

2. The results seem limited to the convex setting.

---

> ### Author Rebuttal · Authors · 2025-03-31
>
> 1. Width-Dominance
>
> Our analysis explicitly relies on the width-dominance regime (Assumption 3.2)​ ensuring as width grows, certain terms dominate the learning dynamics. We will revise to explicitly mention it in the abs/intro. Note that it is a standard condition formalizing the intuition that we operate in the large-width limit where 𝜇P theory is intended to apply. Prior 𝜇P literature also shows partial validity beyond purely “infinite-width” conditions. Besides, our additional experiments in reply to Reviewer g228 show that scale separation behaviors indeed happen in training MLPs.
>
> 2. Justifying Assumption 3.4
>
> Wide-network studies provide strong empirical and theoretical support for A3.4. Prior 𝜇P work (e.g., Yang et al. 2021; Lingle 2024) shows that layer-wise statistics, e.g., activation variances and gradient norms, stabilize early in training and remain nearly invariant across widths. This is consistent with infinite-width theories where macro-level descriptors concentrate, as common in NTK and mean-field analyses. Additionally, recent large-scale experiments (e.g., Vyas et al., 2023; Cerebras “Practitioner’s Guide to 𝜇P”) also confirm that optimal HPs stay stable across model scales, indirectly verifying the training dynamics exhibit self-similarity.
>
> 3. Convex Settings & Strong Convexity of 𝑀(𝜂;𝑛)?
>
> We clarify that our main results do not require global convexity of the loss. Rather, we assume 𝐿-Lipschitz gradients in the standard sense of gradient-based methods (A4.2), typically used in nonconvex analyses to control the norm of the gradients and Hessians locally. Nowhere do we require the global objective to be strictly convex.
>
> In Lemma 4.8, we consider 𝑀(𝜂;𝑛) as the effective scalar function describing the expected loss after training steps 𝑇 under a macro-level parameter 𝜂. For wide enough networks, macro-level fluctuations vanish, and local expansions around 𝜂*(𝑛) ensures a strongly convex local basin in 𝜂. This is akin to standard “strict local minimum” conditions in practical large-scale NNs near an optimal LR. We will clarify in the revision that this does not imply the full model’s loss surface is globally convex.
>
> 4. Novelty and Paper Structure
>
> Our main innovation is the macro–micro separation framework under 𝜇P. Specifically, we derive a continuous-time SDE approximation (Sec. 4) that rigorously separates the convergence of macro-level variables at O(1/𝑛) from the slower micro-level O(1/√𝑛) behavior. This interplay had not been previously formalized; we will state more explicitly.
>
> We will add a clear roadmap of the paper by end of Sec. 2:
>
> Sec. 3 formulates the assumptions and “early-stage coarse-graining” argument.
>
> Sec. 4 establishes the main SDE-based theorems (macro/micro separation).
>
> Secs 5 and 6 apply our unified theoretical framework to explain two empirical phenomenon, on HP and parameter early stability respectively:
>
> Sec. 5 treats the ‘learning rate delay’ using macro-level integrated LR descriptors to explain the lag law;
>
> Sec. 6 extends to the early stabilization of “critical parameter subset”.
>
> 5. Comparison to Xie et al. 2024
>
> Ours and Xie et al. (2024) both leverage SDE to study training dynamics, but from different perspectives. Xie et al predict training loss and optimal LR schedules, deriving convergence rates and escape probabilities under time‑inhomogeneous SDEs, providing an empirical scaling law for HPs. In contrast, we rigorously develop a width scaling theory within μP. We prove a separation of scales between macro-level dynamics (which govern optimal HP transfer) and micro-level fluctuations, thereby explaining why HPs become width‑invariant as the network grows. In short, while Xie et al. offer complementary SDE-based insights on HP sensitivity in non-convex regimes, our work uniquely addresses the underlying mechanism of HP transfer via width-dominance—a question left open in their work.
>
> Technically, Xie et al. is formulated for general non-convex optimization at the cost of ignoring width-specific dynamics​. Our analysis is based on conditions that allow us to rigorously prove separations, typically invoking local convexity or smoothness in the effective (meta-)objective governing HP selection. Empirically, the HP transfer phenomena appears robust even in non-convex NNs. Nonetheless, to extend our rigorous analysis fully to the general non-convex case, additional work would be needed to overcome challenges from multiple local minima and non-unique optimal HPs. We will clarify this comparison/limitation.
>
> 6. Miscellaneous
>
> Citing arXivs: We will update our reference list to cite arXiv's accepted versions whenever available. Our result does not depend on Lingle (2024); the citation was meant to give context of validating and extending 𝜇P in practice.
>
> Generality of Theorem 3.1: You are right. These assumptions are more about how the model is parameterized and how global descriptors converge rather than about specific architectures.

---

### Decision · Program_Chairs · 2025-05-01

**Decision:**

Accept (poster)

**Comment:**

The authors explained why hyperparameter tuning can be done effectively at early stages of training or narrower networks under the Maximal Update Parameterization scheme. They defined the seperation of scales for μP between macro-variable (loss, activation variance, gradient norms etc) and micro-variable (weight values), and proved that macro-variables converge faster than micro-variables. The authors further applied the seperation framework to explain two empirical observations: learning rate scaling laws and delay phenomena, and the "early bird ticket" phenomena (where small subnetworks can be trained to achieve same accuracy as original network).

This paper has many strengths. For example, the paper is well written, with clear definitions of math notations and assumptions. It is good that the authors provide interpretations or implications throughout the paper to help the readers understand the theorems and assumptions. It's interesting that the macro-micro separation framework can be used to explain two phenomenas that seems to be distinct, which widens the impact of this theoretical framework. The claims in the introduction are well-supported and justified. The main theoretical claim is a novel form of scale separation under μP, with key results summarized in Theorem 4.1 and subsequent corollaries.

While the reviewers had some concerns about theoretical claims, the authors did a particularly good job in their rebuttal. Therefore, most of us have agreed to accept this paper for publication! Please include the additional discussion in the next version.